# Inequalities in cardiovascular risks among Swedish adolescents (ABIS): a prospective cohort study

Pär Andersson White ,[1,2] Johnny Ludvigsson,[3] Michael P Jones,[4] Tomas Faresjo[1]

[1]Department of Medicine and Health, Linköping University, Linköping, Sweden
[2]Crown Princess Victoria Children's Hospital, Linköping University Hospital, Linkoping, Sweden
[3]Crown Princess Victoria Children's Hospital and Div. of Pediatrics, Dept. of Clinical and Experimental Medicine, Linkopings universitet, Linkoping, Sweden
[4]Psychology Department, Macquarie University, Sydney, New South Wales, Australia

**Correspondence to**
Dr Pär Andersson White;
par.andersson.white@liu.se

## ABSTRACT

**Objectives** To investigate if socioeconomic status (SES) is predictive of cardiovascular risk factors among Swedish adolescents. Identify the most important SES variable for the development of each cardiovascular risk factor. Investigate at what age SES inequality in overweight and obesity occurs.

**Design** Longitudinal follow-up of a prospective birth cohort.

**Setting** All Babies in Southeast Sweden (ABIS) study includes data from children born between October 1997 and October 1999 in five counties of south east Sweden.

**Participants** A regional ABIS-study subsample from three major cities of the region n=298 adolescents aged 16–18 years, and prospective data from the whole ABIS cohort for overweight and obesity status at the ages 2, 5, 8 and 12 years (n=2998–7925).

**Outcome measures** Blood pressure above the hypertension limit, overweight/obesity according to the International Obesity Task Force definition, low high-density lipoproteins (HDL) or borderline-high low-density lipoproteins according to National Cholesterol Education Program expert panel on cholesterol levels in children.

**Results** For three out of four cardiovascular risk outcomes (elevated blood pressure, low HDL and overweight/obesity), there were increased risk in one or more of the low SES groups (p<0.05). The best predictor was parental occupational class (Swedish socioeconomic classification index) for elevated blood pressure (area under the receiver operating characteristic (ROC) curve 0.623), maternal educational level for overweight (area under the ROC curve 0.641) and blue-collar city of residence for low HDL (area under the ROC curve 0.641). SES-related differences in overweight/obesity were found at age 2, 5 and 12 and for obesity at age 2, 5, 8 and 12 years (all p<0.05).

**Conclusions** Even in a welfare state like Sweden, SES inequalities in cardiovascular risks are evident already in childhood and adolescence. Intervention programmes to reduce cardiovascular risk based on social inequality should start early in life.

## INTRODUCTION

Cardiovascular disease in adults has been shown to be one of the disease groups most correlated with socioeconomic inequalities in the world,[1] and this has also been evident in Swedish adults.[2] Longitudinal studies have

**Strengths and limitations of this study**

► Data derived from a large prospective cohort representative for the general population.
► A standardised procedure for collection of clinical data.
► The Swedish National registers give us valid and reliable data on household income.
► The limited sample size of the follow-up group gives some restraints.

shown that risk factors for cardiovascular disease develop in childhood and that early signs of cardiovascular disease risks are visible in early adulthood.[3–5] The risk factors associated with cardiovascular disease include blood pressure, dyslipidaemia and overweight. Of these risk factors, overweight and obesity are the ones that progress most consistently from childhood to adulthood.[6] Adolescent overweight has been shown to be a stronger predictor of cardiovascular mortality in adulthood than adult overweight.[7]

Recent studies report lower scores than ideal in a range of cardiovascular health factors (blood pressure, cholesterol and glucose level below the American heart association's cut-offs) in European adolescents with low maternal education.[8] In Korea, high income was protective against cardiovascular disease risk factors and in Finland, low socioeconomic status (SES) measured by occupation of the father was associated with higher low-density lipoprotein (LDL) levels in boys and higher systolic blood pressure and higher percentage of smokers.[9 10] However, longitudinal data on the relationships between childhood SES and adolescent's cardiovascular health are still scarce.

An increase in physical activity has been shown to reduce the risk of being overweight and obese as a youth in Sweden and other European countries.[11] Studying the effect of diet on development of overweight and

BMJ

obesity in children and adolescents has yielded no clear conclusions on whether any specific foods are important for the development of obesity.[12] However, there is some evidence for intake of certain foods and development of adult cardiovascular disease. The INTERHEART study found a positive association between intake of meat, salty foods, fried foods and development of cardiovascular disease and a negative association between intake of fruits and vegetables and cardiovascular disease risk.[13]

Sweden currently ranks as ninth in the Organization for Economic Co-operation and Development (OECD) list of economic equality.[14] It has a comprehensive public health system with a number of interventions aimed at reducing social inequalities. These include universal and public financed healthcare equally accessible for all. The healthcare system provides free antenatal consulting and check-ups at Child Health Care Centers which covers advice on nutrition and health from professional nurses for all children at regular intervals up to school age. Other interventions include school-health check-ups and mandatory physical activity in schools and free school lunches for all children. Other support for families with children includes paid maternity leave for 480 days and heavily subsidised daycare.[15 16]

### Aims
The aims of this study were to determine whether SES is predictive of cardiovascular risk factors in Swedish adolescents and to identify which socioeconomic factors that were most important for the development of each risk factor. Further, using longitudinal data, we aimed to shed light on at what ages preventive measures might be important for reducing SES differences in overweight and obesity.

## MATERIAL AND METHODS
### Study population
The All Babies in Southeast Sweden (ABIS) study is a prospective population-based cohort study, initially including 17 055 children born between 1 October 1997 and 1 October 1999 in southeast Sweden in the counties of Östergötland, Jönköping, Kronoberg, Kalmar and Blekinge. The ABIS region is inhabited by approximately 1.25 million out of Sweden's 10 million inhabitants.[17] The region incorporates three of the biggest cities of the country, Linköping (5th), Norrköping (9th) and Jönköping (10th) and also large rural areas, both farmland and forest areas. In terms of income inequality, the counties of southeast Sweden are placed close to the average of the country between the most unequal areas (the capital Stockholm) and the northern areas with the lowest income inequality.[18] This cohort of children has been followed from birth and are still being followed with respect to biological samples, questionnaires and register data. The initial aim of the ABIS study was to analyse the development of immune-mediated diseases including autoimmune diseases, type 1 and type 2 diabetes as well

as cardiovascular disease, and the effect of different environmental and genetic factors.

We undertook a follow-up of a regional subsample of ABIS children and also made a questionnaire-based longitudinal analysis for the whole cohort. Inclusion criteria for the regional subsample was residence in one of the three major cities of Östergötland County. Out of n=578 children invited n=298 (51.6%) participated. The main reasons for non-participation among the invited sample were practical, that is, difficulties to arrange health check-ups for our research nurse at some schools that lacked school-nurse facilities. The study participants filled out a questionnaire about their lifestyle, diet and physical activity. A specially trained research nurse undertook all data collection including blood pressure measurement, questionnaire and collection of blood samples and measurement of anthropometric data.

### Blood pressure
Systolic and diastolic blood pressure was measured manually by the research nurse at the participant's school nursing offices. Study participants rested on a bed at least 5 min before measurement, two consecutive measurements of blood pressure were then taken in the sitting position and the mean value was calculated. Blood pressures were grouped as hypertensive or normotensive according to the guidelines described in the fourth report on the diagnosis, evaluation and treatment of high blood pressure in children and adolescents.[19] These guidelines define hypertension as an average systolic and/or diastolic blood pressure ≥95th percentile for gender, age and height percentile according to the Center of Disease Control and Preventions growth charts on three or more occasions.[20]

### Overweight/obesity
The participating adolescents' height (cm) and weight (kg) were measured by the research nurse. Data on weight and height were missing for eight participants. Body mass index (BMI) was calculated as BMI=weight $(kg)/height (m)^2$ and dichotomised as overweight/obese or not, using the International Obesity Task Force latest age and gender specific cut-offs defined by Cole et al.[21 22] The same procedure was then used for parental-reported height and weight of the child available from previous ABIS follow-ups at age 2, 5, 8 and 12 years.

### Dyslipidaemia
Blood samples for blood lipid analysis were collected from 242 (81.2 %) individuals and analysed for plasma levels of LDL and high-density lipoproteins (HDL) at the Department of Clinical Chemistry, Linköping University Hospital, according to accredited procedures. One sample was missing data on LDL after analysis. Test results were converted from SI units to mg/dL. Results were then grouped as normal or low HDL cholesterol in accordance with the cut-off for low HDL cholesterol (<40 mg/dL) set by the Expert panel on integrated guidelines for

cardiovascular health and risk reduction in children, which is based on the 10th percentile found by the National Cholesterol Educational Program (NCEP).[6] [23] For LDL, only 8 (3.3%) out of 241 had a level above the limit of high LDL (the 95th percentile in NCEP). As this number was insufficient for statistical analyses, the cut-off of borderline high LDL (>110 mg/dL, the 75th percentile in NCEP) was chosen as indicator of increased LDL.

## Socioeconomic variables

A five grade Socioeconomic Class (Swedish socioeconomic classification index (SEI)) variable was constructed based on parents occupation and the occupational groups used by Statistics Sweden in their Swedish socioeconomic classification index.[24] The definitions for each class was: Class I Professionals (SEI 54–60), Class II Intermediate non-manual employees (SEI 44–46), Class III Assistant non-manual employees (SEI 33–36), Class IV Skilled workers (SEI 21–22) and Class V Unskilled workers (SEI 11–12). Occupations in SEI 76–87 which includes non-professional self-employees including farmers were given a Class I–V based on their educational attainment. The adolescents were classified according to the highest occupational class of either of their parents. Mother's educational level was derived from the ABIS birth questionnaire and grouped as primary/secondary education or postsecondary education/university. Family disposable income was derived from Statistics Sweden for the year 2012 and grouped into three groups: low income representing the bottom 20% of household disposable incomes in the original cohort (available for n=16 196), middle income representing the middle 20–80 percentiles of income and top income representing the top 20% of income.

The three included cities in this study represent different social history and different socioeconomic environments.[25] The participants from the city of Motala and Norrköping are towns with a mainly industrial history and were merged in the analysis and labelled as blue-collar cities. The city of Linköping has a history as a political and ecclesiastical administrative centre of the county and was labelled as white-collar city. Today, this city has a workforce of mainly civil servants including high tech industries and one of Sweden's major universities. Previous studies of these cities has exhibit marked public health differences, including higher mortality in cardiovascular disease among adults in the blue-collar cities and difference in life-expectancy.[26] [27]

## Covariates

Age was calculated using the date of answering the questionnaire and the participants birth date. Height was measured at follow-up by a research nurse. Ethnicity was based on the question 'Are you (the mother) born in Sweden' and 'Is the father born in Sweden'. The adolescents were then grouped as having foreign ethnicity if one of the parents was born outside of Sweden. A cardiovascular risk diet index (CRD) was constructed based on the dietary components most associated with myocardial infarction in the INTERHEART study.[13] These factors were: intake of salty foods one time per day or more, intake of deep-fried foods three times or more per week, intake of meat two or more times per day, intake of fruit less than once per day and intake of vegetables less than once per day. This score which has a range of 0–5 was then dichotomised into two groups Low CRD (0–1 items, n=101) and high CRD (2–5 items, n=138). Degree of physical activity was measured by the question 'How many times per week do you do physical activity at least 30 min that is strenuous enough to make you sweat'. The answers were then grouped into two groups; 0–2 times a week, >2 times a week.

## Patient and public involvement

Patients and public were not involved in the design of this study. The results will be disseminated to the study participants through the ABIS study homepage https://www.abis-studien.se/.

## Statistical analysis

Data were analysed using bivariate logistic regression for each cardiovascular risk factor. As analysis of missing values showed significant differences due to SES, adolescents with low maternal educational level had a significantly higher number of missing values in questionnaire data, multiple imputation was used for missing values to reduce this potential bias. Multiple imputation was employed as implemented in the SPSS statistics programme. A Markov Chain Monte Carlo was employed with predictive models selected according to the measurement scale of each dependent variable specified in five imputation samples. A univariate logistic regression was then analysed for each independent variable (SES variables and covariates) and OR with 95% CI for each dependent variable (elevated blood pressure, low-HDL, high-LDL and overweight/obesity at last follow-up) were calculated. For further multivariate analysis, all four SES variables (maternal education, income, SEI, white/blue collar city) were included on a priori basis while other independent variables were chosen based on reaching statistical significance $p \leq 0.05$ or an effect size of OR <0.5 or >2.0 for categorical variables. For continuous variables, height and age were standardised and the OR for 1 SD increase of <0.75 or >1.5 was used as selection criteria. This approach was adopted due to limited sample size and, in this context, effect size being as important from a public health perspective as statistical significance.

The degree of discrimination afforded by each socioeconomic variable was quantified by the area under the receiver operating characteristic (ROC) curve defined by taking the logistic model predicted probability of adverse health state as the test measure and the actual state as an imperfect reference standard. Values >0.5 indicate some discrimination with values close to 1.0 indicating excellent discrimination.

A hierarchical approach was adopted in which SES variables were first adjusted for other SES variables, then

for other independent variables reaching the selection criteria and finally in three of the four outcomes (elevated blood pressure, low LDL and high LDL) for adolescent's overweight status.

Bivariate logistic regression was also used in the final analysis on the relationship between overweight/obesity and maternal educational level at four different time points. The first analysis used a cohort approach and included only subjects with complete data from all follow-ups (n=1070). The second analysis used an available sample approach and included all subjects that had data for any specific age (n=2995–7925).

Statistical analysis was performed using Statistical Package for the Social Sciences V.23 (SPSS, Chicago, Illinois, USA).

## Ethical considerations
All participants in the follow-up gave their informed consent as well as their caregivers consent for those below 18 years of age.

## RESULTS
The participants in our follow-up had a mean age of 16.05 (SD 0.9) years (n=136 males, n=162 females). They were distributed between the cities of Linköping n=160 (54%), Norrköping n=91 (30%) and Motala n=47 (16%). Twenty-four (9.7 %) out of 248 were categorised as having foreign ethnicity. The invited subsample was similar to the original ABIS cohort in respect to a number of variables (birth weight, ethnicity, parental overweight and obesity, child overweight at age 2, 8 and 12 years). However, comparison of this ABIS follow-up with the original ABIS cohort showed a higher proportion of mothers with high education and families with high income and a lower proportion with overweight at 5 years. There was also a slight difference (p=0.052) in gender composition with boys being under-represented in the subsample. Non-participation analysis also showed underrepresentation of children with foreign ethnicity, low maternal education and low household income (see online supplementary table 1).

The total number of participants that were classified as having elevated blood pressure was 49/290 (16.9 %) (see table 1). This was based on 34/290 participants (11.7%) with systolic blood pressures above the limit for hypertension and 17/290 (5.9 %) with diastolic blood pressure above the limit. Two individuals were above both systolic and diastolic hypertension limits. An HDL level below 40 mg/dL was found in 30 (12.4 %) out of 242 samples. For LDL, 25/241 (10.4 %) were above the cut-off for borderline-high value (110 mg/dL).

## Elevated blood pressure
All SES variables were associated with increased ORs for elevated blood pressure in the lowest SES groups compared with the highest (see table 2). Out of the four SES variables, the parental occupational class variable

SEI had the highest area under the ROC curve, 0.623 compared with 0.537–0.578 for the other SES variables. The lowest social class had an OR of 4.37 (95% CI 1.49 to 12.81) compared with the highest class.

After adjustments for all other SES variables, physical activity and overweight status the lowest SEI class had an OR for elevated blood pressure of 4.95 (95% CI 1.30 to 18.89) (see table 3).

## Dyslipidaemia
All indicators of low SES showed an increased odd ratio of having low HDL. Out of the four variables, white/blue-collar city was associated with the highest area under the ROC curve, 0.652 compared with 0.540–0.627 for the other SES variables. Blue-collar city was associated with an OR of 1.78 (95% CI 0.74 to 4.31) of low HDL. Low maternal education and low income were both associated with significantly increased risks of low HDL, low maternal education with an OR of 2.39 (95% CI 1.10 to 5.21) and low income with an OR of 3.94 (95% CI 1.39 to 11.14) for having low HDL.

After adjustment for all other SES variables, gender, CRD and overweight, the OR for blue-collar city was reduced to 1.22 (95% CI 0.46 to 3.27).

For LDL, the difference between low and high SES groups was small, no SES variable reached an OR above 2.0 and no variable reached statistical significance p≤ 0.05. The area under the ROC curve ranged from 0.573 for SEI to 0.511 for white/blue-collar city.

## Overweight/obesity
In the regional subsample, 46/290 (15.8 %) were overweight or obese, with 7 (2.4 %) obese. All four measures of low SES had an increased OR of being overweight or obese. The best discriminator measured as area under the ROC curve for a single SES variable was 0.641 for maternal education. The only other variable that met the criteria for being included in the multivariate analysis was diet according to CRD.

After adjustment for all other SES variables and CRD, maternal education had an OR of 2.39 (95% CI 1.08 to 5.30) for being overweight.

## Overweight and obesity inequality during childhood
Overweight and obesity differences related to maternal education were measured longitudinally at four different time-points. In the analysis on subjects with complete data (cohort approach), the children to low educated mothers had significantly increased odds of being overweight/obese at age 12, but not at age 2, 5 and 8. The analysis based on all available data (available sample approach) showed significant differences at age 2, 5 and 12 but not at age 8. For obesity, there were significant differences at age 5 and 12 in the cohort approach, while the available sample approach showed significant differences at all ages, 2, 5, 8 and 12 years (see table 4).

**Table 1** Characteristics of adolescents by hypertension and overweight status

| | Normal blood pressure | | Elevated blood pressure | | Not overweight | | Overweight | |
|---|---|---|---|---|---|---|---|---|
| | N | % | N | % | N | % | N | % |
| Total | 240 | 83.0 | 49 | 17.0 | 244 | 84.1 | 46 | 15.9 |
| Sex | | | | | | | | |
| Boy | 105 | 43.8 | 27 | 55.1 | 109 | 44.7 | 24 | 52.2 |
| Girl | 135 | 56.2 | 22 | 44.9 | 135 | 55.3 | 22 | 47.8 |
| Ethnicity | | | | | | | | |
| Swedish | 213 | 89.9 | 44 | 91.7 | 218 | 90.5 | 40 | 88.9 |
| Foreign ethnicity | 24 | 10.1 | 4 | 8.3 | 23 | 9.5 | 5 | 11.1 |
| Physical activity—times per week | | | | | | | | |
| >2 | 138 | 67.6 | 31 | 83.8 | 149 | 72.0 | 21 | 60.0 |
| 0–2 | 66 | 32.4 | 6 | 16.2 | 58 | 28.0 | 14 | 40.0 |
| CRD | | | | | | | | |
| 0–1 item | 88 | 44.4 | 14 | 40.0 | 94 | 46.5 | 8 | 25.0 |
| 2–5 items | 110 | 55.6 | 21 | 60.0 | 108 | 53.5 | 24 | 75.0 |
| BMI group of adolescent | | | | | | | | |
| Not OW | 213 | 88.8 | 31 | 63.3 | | | | |
| OW/OB | 27 | 11.2 | 18 | 36.7 | | | | |
| Income | | | | | | | | |
| High | 94 | 39.5 | 16 | 33.3 | 99 | 40.9 | 12 | 26.7 |
| Middle | 111 | 46.6 | 26 | 54.2 | 112 | 46.3 | 25 | 55.6 |
| Low | 33 | 13.9 | 6 | 12.5 | 31 | 12.8 | 8 | 17.8 |
| Maternal education | | | | | | | | |
| Tertiary | 138 | 58.2 | 20 | 41.7 | 144 | 59.8 | 14 | 31.1 |
| <Tertiary | 99 | 41.8 | 28 | 58.3 | 97 | 40.2 | 31 | 68.9 |
| SEI | | | | | | | | |
| Class I | 59 | 24.6 | 5 | 10.2 | 60 | 24.6 | 4 | 8.7 |
| Class II–IV | 131 | 54.6 | 25 | 51.0 | 130 | 53.3 | 27 | 58.7 |
| Class V | 50 | 20.8 | 19 | 38.8 | 54 | 22.1 | 15 | 32.6 |
| City of residence | | | | | | | | |
| White-collar | 132 | 55 | 23 | 46.9 | 137 | 56.1 | 18 | 39.1 |
| Blue-collar | 108 | 45 | 26 | 53.1 | 107 | 43.9 | 28 | 60.9 |
| | M | SD | M | SD | M | SD | M | SD |
| Age at follow-up (years) | 16.02 | 0.90 | 16.12 | 1.04 | 16.02 | 0.92 | 16.18 | 0.94 |
| Height (m) | 1.72 | 0.09 | 1.73 | 0.10 | 1.73 | 0.09 | 1.71 | 0.09 |

BMI, body mass index; CRD, cardiovascular risk diet index; OW/OB, overweight or obese; SEI, Swedish socioeconomic classification index.

## DISCUSSION

There was significantly increased risk of three out of four cardiovascular risk factors due to low SES in Swedish adolescents. Parental occupational class according to SEI was the most important factor for risk of elevated blood pressure. Although overweight was important for elevated blood pressure, the independent importance of parental occupational class was further strengthened as it remained after adjustment for all other SES factors, physical activity and overweight. For other outcomes, we

found maternal education to be the best discriminator for overweight and white/blue collar city for low HDL (<40 mg/dL). Our results further show that socioeconomic differences in overweight and obesity are of importance in adolescents and early childhood. This finding yields implications for preventive measures and suggests that these measures should start early in life.

In the Whitehall studies, occupational class and blood pressure were found to be closely associated in adults.[28] One finding of the Whitehall II study is that workers with

**Table 2** Univariate analysis with ORs for cardiovascular risk factors and areas under the ROC-curve for SES variables (n=298)

| | Elevated blood pressure | | Low HDL | | High LDL | | Overweight | |
|---|---|---|---|---|---|---|---|---|
| | OR | CI | OR | CI | OR | CI | OR | CI |
| Age at follow-up (Z-years) | 1.12 | 0.83 to 1.51 | 1.12 | 0.79 to 1.59 | 0.99 | 0.63 to 1.53 | 1.19 | 0.87 to 1.63 |
| Height (Z-m) | 1.24 | 0.91 to 1.64 | 1.41 | 0.85 to 2.33 | 1.26 | 0.85 to 1.88 | 0.82 | 0.59 to 1.14 |
| Gender | | | | | | | | |
| Girl | 1.0 | | 1.0 | | 1.0 | | 1.0 | |
| Boy | 1.61 | 0.86 to 3.04 | 2.78 | 1.09 to 7.09 | 1.31 | 0.57 to 3.00 | 1.38 | 0.73 to 2.60 |
| Ethnicity | | | | | | | | |
| Swedish | 1.0 | | 1.0 | | 1.0 | | 1.0 | |
| Foreign | 0.83 | 0.27 to 2.58 | 1.49 | 0.53 to 4.19 | 0.29 | 0.04 to 2.25 | 1.13 | 0.41 to 3.16 |
| Physical activity—times/week | | | | | | | | |
| >2 | 1.0 | | 1.0 | | 1.0 | | 1.0 | |
| 0–2 | 0.39 | 0.15 to 1.05 | 0.93 | 0.41 to 2.26 | 0.68 | 0.28 to 1.67 | 1.60 | 0.72 to 3.55 |
| CRD | | | | | | | | |
| 0–1 item | 1.0 | | 1.0 | | 1.0 | | 1.0 | |
| 2–5 items | 1.12 | 0.49 to 2.61 | 2.41 | 0.95 to 6.15 | 1.41 | 0.63 to 3.16 | 2.80 | 1.06 to 7.41 |
| BMI group of adolescent | | | | | | | | |
| Not OW | 1.0 | | 1.0 | | 1.0 | | | |
| OW/OB | 4.42 | 2.19 to 8.95 | 3.44 | 1.47 to 8.07 | 3.20 | 1.28 to 7.98 | | |
| Income (ROC) | 0.537 | | 0.627 | | 0.545 | | 0.580 | |
| High | 1.0 | | 1.0 | | 1.0 | | 1.0 | |
| Middle | 1.35 | 0.68 to 2.67 | 0.88 | 0.36 to 2.16 | 1.08 | 0.42 to 2.77 | 1.90 | 0.91 to 3.97 |
| Low | 1.11 | 0.41 to 3.05 | 3.94 | 1.39 to 11.14 | 1.73 | 0.46 to 6.54 | 2.21 | 0.82 to 5.95 |
| Maternal education (ROC) | 0.578 | | 0.610 | | 0.564 | | 0.641 | |
| Tertiary | 1.0 | | 1.0 | | 1.0 | | 1.0 | |
| <Tertiary | 1.87 | 0.99 to 3.53 | 2.39 | 1.10 to 5.21 | 1.67 | 0.77 to 3.59 | 3.22 | 1.61 to 6.44 |
| SEI (ROC) | 0.623 | | 0.540 | | 0.573 | | 0.602 | |
| Class I | 1.0 | | 1.0 | | 1.0 | | 1.0 | |
| Class II–IV | 2.20 | 0.80 to 6.05 | 1.33 | 0.48 to 3.66 | 1.91 | 0.60 to 6.09 | 3.21 | 1.08 to 9.57 |
| Class V | 4.37 | 1.49 to 12.81 | 1.45 | 0.48 to 4.48 | 1.41 | 0.24 to 8.28 | 4.28 | 1.34 to 13.73 |
| City of residence (ROC) | 0.541 | | 0.652 | | 0.511 | | 0.585 | |
| White-collar | 1.0 | | 1.0 | | 1.0 | | 1.0 | |
| Blue-collar | 1.39 | 0.76 to 2.56 | 1.78 | 0.74 to 4.31 | 1.10 | 0.51 to 2.41 | 2.00 | 1.06 to 3.78 |

BMI, body mass index; CRD, cardiovascular risk diet index; HDL, high-density lipoproteins; LDL, low-density lipoproteins; OW/OB, overweight or obese; ROC, area under the receiver operating characteristic curve; SEI, Swedish socioeconomic classification index; SES, socioeconomic status.

low occupational grade have increased blood pressure during the working day, but no difference in resting blood pressure.[29] As blood pressure in this study was measured while the adolescents' were attending school, it is possible that it represents the same kind of increase in response to school tasks in low SES adolescents as that seen during the work day in adult workers.

The impact of maternal education on child overweight has been shown in large multicohort studies.[30] In addition to being a marker for SES like occupation and income, the impact of maternal education could involve differences in knowledge. There is evidence that adolescents with mothers with good nutritional knowledge

have a healthier diet.[31] High maternal education has also been associated with a higher degree of breakfast eating in adolescents.[32] Further, low parental education has been associated with increased access to sugar sweetened drinks and low vegetable intake.[33] These factors could be part of the explanation of why maternal educational level is more important than other SES factors for overweight status in adolescents.

White/blue-collar city was the best discriminator of low HDL status. After adjustment for other SES variables the OR for blue-collar cities was reduced to OR 1.35 (0.53 to 3.48), adjusting for diet and overweight further decreased the OR to 1.22 (0.46 to 3.27) indicating that most of the

**Table 3** Multivariate analysis with ORs for cardiovascular risk factors

| | Model 1 | | Model 2 | | Model 3 | |
|---|---|---|---|---|---|---|
| | OR | CI | OR | CI | OR | CI |
| **Elevated blood pressure** | | | | | | |
| Income | | | | | | |
| High | 1.00 | | 1.00 | | 1.00 | |
| Middle | 0.85 | 0.39 to 1.85 | 0.85 | 0.38 to 1.89 | 0.75 | 0.32 to 1.75 |
| Low | 0.60 | 0.20 to 1.81 | 0.69 | 0.22 to 2.12 | 0.60 | 0.17 to 1.96 |
| Maternal education | | | | | | |
| Tertiary | 1.00 | | 1.00 | | 1.00 | |
| <Tertiary | 1.29 | 0.60 to 2.77 | 1.15 | 0.52 to 2.54 | 0.88 | 0.37 to 2.08 |
| SEI | | | | | | |
| Class I | 1.00 | | 1.00 | | 1.00 | |
| Class II–IV | 2.21 | 0.77 to 6.34 | 2.25 | 0.78 to 6.52 | 2.09 | 0.70 to 6.27 |
| Class V | 4.18 | 1.22 to 14.41 | 4.43 | 1.26 to 15.58 | 4.95 | 1.30 to 18.89 |
| City of residence | | | | | | |
| White-collar | 1.00 | | 1.00 | | 1.00 | |
| Blue-collar | 1.12 | 0.58 to 2.14 | 1.18 | 0.61 to 2.30 | 1.05 | 0.52 to 2.13 |
| **Low HDL (<0.40 mg/dL)** | | | | | | |
| Income | | | | | | |
| High | 1.00 | | 1.00 | | 1.00 | |
| Middle | 0.98 | 0.31 to 3.08 | 0.72 | 0.26 to 1.98 | 0.73 | 0.26 to 2.20 |
| Low | 3.16 | 0.97 to 10.35 | 3.28 | 0.92 to 11.67 | 3.41 | 0.89 to 13.06 |
| Maternal education | | | | | | |
| Tertiary | 1.00 | | 1.00 | | 1.00 | |
| <Tertiary | 2.56 | 0.99 to 6.62 | 2.29 | 0.86 to 6.09 | 2.00 | 0.73 to 5.50 |
| SEI | | | | | | |
| Class I | 1.00 | | 1.00 | | 1.00 | |
| Class II–IV | 0.88 | 0.32 to 2.48 | 0.82 | 0.29 to 2.36 | 0.76 | 0.26 to 2.10 |
| Class V | 0.53 | 0.14 to 2.07 | 0.41 | 0.10 to 1.73 | 0.41 | 0.09 to 1.75 |
| City of residence | | | | | | |
| White-collar | 1.00 | | 1.00 | | 1.00 | |
| Blue-collar | 1.41 | 0.55 to 3.60 | 1.35 | 0.53 to 3.48 | 1.22 | 0.46 to 3.27 |
| **High LDL (>110 mg/dL)** | | | | | | |
| Income | | | | | | |
| High | 1.00 | | 1.00 | | 1.00 | |
| Middle | 0.91 | 0.32 to 2.56 | 0.88 | 0.31 to 2.51 | 0.84 | 0.29 to 2.49 |
| Low | 1.48 | 0.35 to 6.21 | 1.47 | 0.34 to 6.29 | 1.41 | 0.31 to 6.39 |
| Maternal education | | | | | | |
| Tertiary | 1.00 | | 1.00 | | 1.00 | |
| <Tertiary | 1.86 | 0.70 to 4.93 | 1.81 | 0.68 to 4.79 | 1.58 | 0.58 to 4.32 |
| SEI | | | | | | |
| Class I | 1.00 | | 1.00 | | 1.00 | |
| Class II–IV | 1.69 | 0.49 to 5.77 | 1.68 | 0.49 to 5.79 | 1.52 | 0.43 to 5.40 |
| Class V | 0.92 | 0.10 to 8.56 | 0.97 | 0.10 to 9.06 | 0.90 | 0.09 to 9.54 |
| City of residence | | | | | | |
| White-collar | 1.00 | | 1.00 | | 1.00 | |

Continued

**Table 3** Continued

| | Model 1 | | Model 2 | | Model 3 | |
|---|---|---|---|---|---|---|
| | OR | CI | OR | CI | OR | CI |
| Blue-collar | 0.89 | 0.37 to 2.15 | 0.90 | 0.37 to 2.18 | 0.83 | 0.33 to 2.09 |
| Overweight status | | | | | | |
| Income | | | | | | |
| High | 1.00 | | 1.00 | | | |
| Middle | 1.16 | 0.52 to 2.60 | 1.12 | 0.50 to 2.54 | | |
| Low | 1.21 | 0.42 to 3.54 | 1.20 | 0.40 to 3.56 | | |
| Maternal education | | | | | | |
| Tertiary | 1.00 | | 1.00 | | | |
| <Tertiary | 2.63 | 1.20 to 5.78 | 2.39 | 1.08 to 5.30 | | |
| SEI | | | | | | |
| Class I | 1.00 | | 1.00 | | | |
| Class II–IV | 2.34 | 0.75 to 7.28 | 2.09 | 0.66 to 6.60 | | |
| Class V | 1.90 | 0.52 to 6.98 | 1.63 | 0.43 to 6.25 | | |
| City of residence | | | | | | |
| White-collar | 1.00 | | 1.00 | | | |
| Blue-collar | 1.39 | 0.70 to 2.73 | 1.39 | 0.70 to 2.76 | | |

Adjusted for SES and other explanatory variables (n=298).
Model 1 includes all SES variables. Model 2 includes all SES and other variables excluding overweight status that reached the selection criteria in univariate analysis (p≤0.05 or OR <0.5 or >2.0 for categorical and <0.75 or >1.5 for continuous variables): physical activity for elevated blood pressure; gender and cardiovascular risk diet for low HDL; ethnicity for LDL; cardiovascular risk diet for overweight. Model 3 includes all SES, other explanatory variables and overweight status.
BMI, body mass index; HDL, high-density lipoproteins; LDL, low-density lipoproteins; SEI, Swedish Socioeconomic Classification Index; SES, socioeconomic status.

**Table 4** Association between maternal education level at birth and child overweight or obesity at four ages

| Time point: | 2 years | 5 years | 8 years | 12 years |
|---|---|---|---|---|
| Cohort approach | | | | |
| Age: mean (SD) | 2.72 (0.28) | 5.32 (0.30) | 7.86 (0.34) | 12.61 (0.87) |
| Sample size available | 1070 | 1070 | 1070 | 1070 |
| Overweight: n (%) | 149 (13.9%) | 146 (13.6%) | 157 (14.7%) | 168 (15.7%) |
| Obese: n (%) | 19 (1.8%) | 27 (2.5%) | 21 (2.0%) | 21 (2.0%) |
| Tertiary vs <Tertiary education: overweight* | 0.95 (0.67 to 1.34) | 1.42 (1.00 to 2.03) | 1.36 (0.96 to 1.92) | 1.81 (1.28 to 2.55) |
| Tertiary vs <Tertiary education: obesity* | 0.41 (0.15 to 1.08) | 3.21 (1.29 to 8.02) | 1.46 (0.60 to 3.56) | 3.89 (1.30 to 11.63) |
| Available sample approach | | | | |
| Age: mean (SD) | 2.73 (0.30) | 5.34 (0.31) | 7.86 (0.38) | 12.40 (0.91) |
| Sample size available | 7925 | 6602 | 2998 | 3337 |
| Overweight: n (%) | 1187 (15.0%) | 1096 (16.6%) | 446 (14.9%) | 527 (15.8%) |
| Obese: n (%) | 191 (2.4%) | 273 (4.1%) | 71 (2.4%) | 75 (2.2%) |
| Tertiary vs <Tertiary education: overweight* | 1.18 (1.03 to 1.34) | 1.30 (1.13 to 1.50) | 1.16 (0.94 to 1.42) | 1.70 (1.39 to 2.06) |
| Tertiary vs <Tertiary education: obesity* | 1.56 (1.12 to 2.17) | 1.52 (1.15 to 1.99) | 2.28 (1.31 to 3.95) | 2.16 (1.29 to 3.61) |

The cohort approach uses only participants with all necessary data recorded at all four ages whereas the available sample approach uses any participant with all necessary data recorded on at least one age.
*OR (95% CI).

risk associated with blue-collar residence is due to differences in SES, diet and overweight. Thus, our results are in line with previous studies.[25]

The main limitation of this study is the limited number of participants of the follow-up part (n=298). Even though the sample size of study is somewhat restrained we managed to capture SES-related significant differences in cardiovascular risks. Our sample is quite representative concerning overweight status compared with the previous larger samples in ABIS and to other Swedish studies.[34] Although the number of low-income families were similar in our invited subsample compared with the original cohort, there was an overrepresentation of high maternal education and high-income families and this most likely reflects the fact that the subsample is from a more urban population than the rest of the ABIS region and must be taken into consideration when interpreting the result. Among the invited, the number of adolescents with high maternal education at birth were higher among the participants of the study than in the non-participant group and the same was true for high-income families. This is a common finding in longitudinal studies, but as previous studies have shown, the effect of this underrepresentation in low SES groups, if anything, is to underestimate inequality slightly in later follow-ups and have little effect on the qualitative conclusion on inequality.[35] The number of adolescents with elevated blood pressure was quite high (16.9%) and as blood pressure was measured only at one visit to school by the research nurse some of the adolescents might have had White-coat rather than true hypertension.[19] However, other studies have shown similarly high values in other populations.[36] A recent study in a Swedish cohort also found higher average systolic blood pressure in the age group 14–16 years than was found in the fourth report on the diagnosis, evaluation and treatment of high blood pressure in children and adolescents.[37] We could not rule out that White-coat hypertension might be related to socioeconomic factors but have no evidence of that.

The fact that the longitudinal data were based on parental-reported height and weight might affect the accuracy of our results. A German study found that parental reported data tend to overestimate the prevalence of overweight in young children and underestimate it in older children; however, the same study found the impact of SES on this error to be small.[38] Thus, although the exact prevalence of overweight status in our study might differ slightly from the true prevalence, the relative difference between SES groups should not.

Different socioeconomic variables can have different effects on health. It is a strength that we investigated not one but four dimensions of SES inequality. These dimensions seem to be of varying importance for cardiovascular risk outcomes. Thus, our study is an example of the many layers of the social determinants of health that give rise to an individual's risk of disease.[39] Although income, maternal education and occupational status are correlated, it is not surprising that the strongest discriminators for cardiovascular risk factors are occupational class, blue collar city and educational level rather than income in this Swedish cohort, as income inequality still is low compared with most other countries.[40]

The implication of adolescent SES differences on adult life cardiovascular health inequalities could be important. Longitudinal data from Finland have shown that cardiovascular risk factors such as elevated blood pressure, serum lipids and obesity track to adulthood, increasing the risk of adult abnormalities.[41] Data from the same cohort also show that adolescent blood pressure and LDL cholesterol are risk factors of adult carotid intima media thickening independent of adult risk factors.[42] A general awareness in primary care is important of the potential risk of elevated blood pressure and other cardiovascular risks in low SES adolescents.

Sweden has for a long time been considered one of the more egalitarian countries in the world and has often been used for comparison in studies on health inequalities as a good example.[43 44] Yet during the last two decades, this has started to change with economic inequality increasing faster than in other OECD countries and Sweden has gone from a top position in equality to now being the ninth country in income equality out of the 36 countries in OECD.[14] Previous studies have found that for health problems with higher prevalence in low SES groups, like cardiovascular disease in adults, both the overall prevalence and absolute inequality between high and low SES groups increases with larger economic inequality.[1 45] Our finding that cardiovascular risks in children and adolescents have a SES gradient also in Sweden could have serious public health implication for the future, as the rapid rise of economic inequality might increase absolute inequalities and prevalence of cardiovascular risk factors in children and adolescents. Concerns for health inequalities have increased in recent years spurring the Swedish government to start a Commission for health equality with the aim of finding ways to decrease these inequalities.[46]

## CONCLUSION

Despite an extensive welfare system and public health interventions in Sweden, cardiovascular health inequality is evident among Swedish youths and children. Preventive health services and clinicians must be aware of the increased risk of cardiovascular risk factors in low SES children and adolescents and improve prevention programmes in this group. Policies aimed at decreasing the evident inequalities in cardiovascular disease in adults should start early and include interventions already in childhood and adolescence.

**Contributors** TF, JL and PAW were responsible for the study concept and design. JL is the principal investigator for ABIS. PAW did the statistical analysis with advice from MPJ. All authors were responsible for analysis and interpretation of data. PAW drafted the manuscript and TF, JL and MPJ undertook critical revision of the manuscript. PAW has full access to all of the data in the study, takes responsibility for the integrity of the data and accuracy of the data analysis and had the final responsibility to submit for publication.

**Funding** The ABIS-study has been supported by Swedish Research Council (K2005-72X-11242-11A and K2008-69X-20826-01-4) and the Swedish Child Diabetes Foundation (Barndiabetesfonden), JDRF Wallenberg Foundation (K 98-99D-12813-01A), Medical Research Council of Southeast Sweden (FORSS) and the Swedish Council for Working Life and Social Research (FAS2004–1775) and Östgöta Brandstodsbolag. The Research and PhD studies Committee (FUN), Linköping University, Sweden (LiU-).

**Competing interests** None declared.

**Patient consent for publication** Not required.

**Ethics approval** The ABIS-study has been approved by the Research Ethics committees of the Faculty of Health Science at Linköping University and Lund university. Dnr 03–092, Dnr 99–227 and Dnr 99–321.

**Provenance and peer review** Not commissioned; externally peer reviewed.

**Data availability statement** Data are available on reasonable request. Data from the ABIS registry, including study protocols, report forms and consents may be obtained through requests to the PI and project coordinator Senior professor Johnny Ludvigsson, email: Johnny.Ludvigsson@liu.se.

**ORCID iD**
Pär Andersson White http://orcid.org/0000-0003-2963-6507

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
