## [Reviewer comments · BMJ Open]

ARTICLE DETAILS

TITLE (PROVISIONAL)	Inequalities in Cardiovascular risks among Swedish adolescents – ABIS a prospective cohort study
AUTHORS	Andersson White, Pär; Ludvigsson, Johnny; Jones, Michael; Faresjo, Tomas

VERSION 1 - REVIEW

REVIEWER	Roya Kelishadi Isfahan University of Medical Sciences, Iran
REVIEW RETURNED	22-Jul-2019

GENERAL COMMENTS	-The introduction should be summarized.-More information should be provided on the study population. It is important for international readers.-The conclusion should be more concise and precise.
--

REVIEWER	Yipu Shi Public Health Agency of Canada
REVIEW RETURNED	12-Sep-2019

GENERAL COMMENTS	The study examined associations of various socioeconomic (SES) measures with cardiovascular risk (CVD) in a Sweden cohort followed from birth to adolescent years. Despite the many strengths of the data used for this study (population-based, objectively measured CVD and SES variables etc.), the sample size was small, which limited a reliable interpretation of the results, especially as authors have acknowledged that a higher participation from the high income and high maternal education families of Sweden adolescents. This limitation is clearly evidenced by the small number of counts in Table 1 and large confidence intervals in the regression analysis in (Table 2 and 3). Some minor questions: - Page 9 line 43, it is not clear how the Swedish socioeconomic classification index (SEI) is defined. It seems like a composite score for income and maternal education combined. Do you need SEI when you have income and maternal education reported separately.- Page 10 line 40, Multiple imputation was used to preserve sample size, please state what imputation method was used.
--

REVIEWER	Rema Ramakrishnan University of Oxford, UK
REVIEW RETURNED	13-Oct-2019

GENERAL COMMENTS	The authors examined socioeconomic differences in cardiovascular risk factors, aimed to identify which of the socioeconomic factors were important for each risk factor, and also assessed at what age socioeconomic differences in overweight and obesity occurred during childhood. This is an interesting topic and the dataset used is appropriate for this study. If the writing could be improved then perhaps it would have been easier to comprehend the information presented in the paper. Additionally, the methods' section needs a lot of work. ABSTRACT: The objectives need to be refined to include what the authors have stated in the introduction of the paper. The results' section needs to have estimates and their associated confidence intervals. INTRODUCTION Page 4: Lines 22-25: The reference cited does not conclude that out of all the risk factors overweight and obesity are the ones that progress most consistently from childhood to adulthood. The paper referred to concluded that excess weight in adolescence persisted into adulthood and that cardiovascular risk factors tracked strongly in the overweight participants compared to the lean participants. Page 4: Lines 11-16: The information here is a bit repetitive and is very similar to that described in paragraph 2 of the introduction. Page 5: Lines 17-46: The justification given here for the study is bit confusing. Based on the information provided here it would have been justifiable to compare the impact of SES on cardiovascular risk factors in the 1980s to that in 2016 and after that. METHODS Page 6: Lines 43-46: Not sure why only participants from one county (out of the six counties in the original study) was included in this study. Page 6: Lines 48-59 and Page 7: Lines 3-21: This description should probably be in the results section. Page 7: Lines 3- 14: It looks like the analysis is for nonparticipation and not drop out. Page 7: Lines 21-31: This should be in the description of socioeconomic variables' section. The heading, questionnaire, could be changed to, diet and physical activity, because these were the only information described in this section. Page 8: Blood pressure: The authors have described that the blood pressure was categorised into hypertensive and normotensive as per the 4th report of the diagnosis, evaluation, and treatment of high blood pressure in children and adolescents. It would be informative to have the values that define hyper- and normotensive adolescents (As per the report hypertension is defined as average SBP and/or diastolic BP (DBP) that is ≥ 95th
---

percentile for gender, age, and height on ≥ 3 occasions). Also, compared to this definition, the current study measured BP on only two occasions.

Page 8: The section on anthropometrical factors can be renamed as body mass index.

How were weight and height assessed? Was it self-reported or was it measured by the research staff? (On reading the discussion it seems these were self-reported by the parents, but it would be helpful to have this information here). This is important to know since it can affect how the results are interpreted.

Page 8: Lines 48-51: It should be BMI calculated as ... and not Obesity and BMI calculated as.....

Page 8: Lines 54-56: Cutoff based on what?

Page 9: Lines 3-6: Not sure why parental height was included?

Page 9: Lines 32-35: Was the definition of borderline high LDL also based on percentiles? If yes, what percentile was used and if no, what was this definition based on?

Not sure why triglycerides was not included in the measurement for dyslipidaemia. It would have been useful to include this variable.

Page 9: Socioeconomic variables: The citation given on line 42-43 for the Swedish socioeconomic classification index appears to be incorrect. Though the authors have described that parental occupation was included and maternal education and family income were measured it is unclear how the authors constructed this index in the current study and what the five grades were. Also, from the analysis and results it seems that SES was assessed not only by this index but also from individual SES variables. It would be informative to state that here in this section for clarity. What does each class include and what does it mean for SES?

Page 10: The white collar and blue collar city section could possibly be included in the section on socioeconomic variables and not have a separate section for this variable because white/blue collar cities primarily refer to SES.

Page 10: Potential confounders:

It would be informative to describe how birth weight was assessed – continuous or categorical (the number and description of the categories). However, it is unclear how birthweight is a confounder because it is most probably a mediator. The SES can affect birthweight which can be a risk factor for the cardiovascular risk factors.

Page 10: Statistical analysis:

It is important to describe for each cardiovascular risk factor at which time they were measured. BMI was measured at four time points but it's unclear when the other risk factors were measured. Also, out of the four measurements for BMI which one was used to examine the overall association between SES and overweight/obesity?

It is unclear what the independent variables and potential confounders are. It seems the confounders have been treated as independent variables. It would help to have consistent terms. Based on the research questions/aims of the study the SES variables are the independent variables and cardiovascular risk factors (blood pressure, dyslipidaemias, and overweight) are the dependent variables/outcomes.

The current study is a prospective cohort study and the outcome is not rare. Therefore it is unclear why logistic regression was used to estimate odds ratios. It would have been more appropriate to compute relative risk or risk ratio.

It would be informative to include what multiple imputation method was used and how many datasets were used for imputation.

Page 10: Lines 56-60: It is unclear what the four SES variables are (though from table 1 it seems that these are: maternal education, income, SEI, and blue collar/white collar cities). It would be informative to make this clear in the section on socioeconomic variables. It is also unclear what the other independent variables are (though the authors say that the continuous variables are height, age, and birthweight). The variable age was not mentioned in the section on potential confounders. If BMI was one of the outcomes then including height as an independent variable/confounder would be incorrect.

Page 11: Lines 6-13: It is interesting to read that effect size was used to determine inclusion of variables in the model. It would be informative to provide a citation as to why the authors decided to use <0.75 or >1.5 as selection criteria to include the variables. Also, on page 10: lines 59-60, the authors have described that $p \leq 0.05$ or the effect size of $OR < 0.5$ or > 2.0 was used for categorical variables. However, it is unclear in which situation either was used.

Page 11: 29-32: The SES variables are likely to be correlated with each other and there is possibility for multicollinearity that can affect the estimates especially the variance. How did the authors take this into account especially when estimating the odds ratios? Also, it would be informative to list the lifestyle factors because it was not mentioned what these refer to (though it would seem that diet and physical activity). It is unclear why SES should be adjusted for overweight status if this variable is one of the cardiovascular risk factors and hence one of the outcomes/dependent variables.

Page 11: Lines 34-46: It would have been more appropriate to state that logistic regression was used to examine the relationship between maternal education at four different time points and overweight/obesity because this is cohort study and not a case-control study. Also, it's unclear why only maternal education was used for this purpose and not other SES variables. The study aim was to determine at what age socioeconomic differences in overweight and obesity occur....The measurements at four time points are likely to be correlated. How did the logistic regression model account for this at each time point? The methodology used does not seem to be satisfactory for answering the question, what age socioeconomic differences in overweight and obesity occur...?

	It would be informative to examine/describe the characteristics of the cities that were designated as blue and white collar cities and have a descriptive table for it. The authors have used ROC to quantify the degree of discrimination. It would be informative to have analysis for calibration also and corresponding calibration plots too. RESULTS The authors have compared ROCs, however, no statistical testing was done if the differences were statistically significant or not. It would have been easier if the descriptive statistics were presented first and then the inferential statistics instead of describing the statistics by each variable. Page 12: Lines 22-28: This information could possibly be deleted. Page 12: Lines 42-48: This information is redundant because these variables are potential confounders and as such not the study aims. Page 13: Lines 3-6: It is unclear what indicators of low SES the authors are talking about. Page 13: Lines 23-33: As mentioned above for blood pressure, these variables are not the focus of the research question (they are potential confounders) and hence their estimates are redundant. Page 13: The subtitle should possibly be overweight/obese because the authors combined these two into a single category. Page 14: Lines 3-6: Lifestyle factors (diet, physical activity) were not included in the aims of the study. They are potential confounders. Table 2: The study aims relate to SES and cardiovascular risk factors and therefore it would be appropriate to include statistics for these variables only and not for other variables listed in the table. Also, it is unclear why height and overweight should be included as potential confounders when being overweight/obese is an outcome.
--	--

VERSION 1 – AUTHOR RESPONSE

Reviewer 1

Please leave your comments for the authors below

-The introduction should be summarized.

The introduction has been summarized to some extent

-More information should be provided on the study population. It is important for international readers.

We have added information about the ABIS region in study population on page 6.

-The conclusion should be more concise and precise.

Conclusion has been rewritten

Reviewer 2

Page 9 line 43, it is not clear how the Swedish socioeconomic classification index (SEI) is defined. It seems like a composite score for income and maternal education combined. Do you need SEI when you have income and maternal education reported separately.

Clarification regarding construction of the parental occupational class/SEI variable has been added, see page 9.

- Page 10 line 40, Multiple imputation was used to preserve sample size, please state what imputation method was used.

Information about Multiple imputation method is added on page 11

Reviewer 3

ABSTRACT:

The objectives need to be refined to include what the authors have stated in the introduction of the paper.

Objectives have been rewritten in the abstract.

The results' section needs to have estimates and their associated confidence intervals.

Estimates have been added.

INTRODUCTION

Page 4: Lines 22-25: The reference cited does not conclude that out of all the risk factors overweight and obesity are the ones that progress most consistently from childhood to adulthood. The paper referred to concluded that excess weight in adolescence persisted into adulthood and that cardiovascular risk factors tracked strongly in the overweight participants compared to the lean participants.

This reference has been changed to

Expert Panel on Integrated Guidelines for Cardiovascular H, Risk Reduction in C, Adolescents, et al. Expert panel on integrated guidelines for cardiovascular health and risk reduction in children and adolescents: summary report. Pediatrics 2011;128 Suppl 5:S213-56. doi: 10.1542/peds.2009-2107C

Page 4: Lines 11-16: The information here is a bit repetitive and is very similar to that described in paragraph 2 of the introduction.

The repetitive lines have been removed.

Page 5: Lines 17-46: The justification given here for the study is bit confusing. Based on the information provided here it would have been justifiable to compare the impact of SES on cardiovascular risk factors in the 1980s to that in 2016 and after that.

We believe that this paragraph is a necessary description of the current situation in Sweden, that is increasing income inequality (that some reader might not be aware of) but with a lot of policies that should be protective against health inequalities in children and adolescents.

METHODS

Page 6: Lines 43-46: Not sure why only participants from one county (out of the six counties in the original study) was included in this study.

Participation was limited to Östergötland County to limit travel time and resources needed for the research nurses involved in this study. Also, information on participants of ABIS in Östergötland is more complete than for other counties as some previous papers have been limited to this part of the cohort. Ref

Page 6: Lines 48-59 and Page 7: Lines 3-21: This description should probably be in the results section.

Description of the subsample moved to result.

Page 7: Lines 3- 14: It looks like the analysis is for nonparticipation and not drop out.

We have changed this to Non-participation.

Page 7: Lines 21-31: This should be in the description of socioeconomic variables' section.

We agree; these lines have been moved to SES variables

The heading, questionnaire, could be changed to, diet and physical activity, because these were the only information described in this section.

Heading have been changed as suggested.

Page 8: Blood pressure: The authors have described that the blood pressure was categorised into hypertensive and normotensive as per the 4th report of the diagnosis, evaluation, and treatment of high blood pressure in children and adolescents. It would be informative to have the values that define hyper- and normotensive adolescents (As per the report hypertension is defined as average SBP and/or diastolic BP (DBP) that is ≥ 95 th percentile for gender, age, and height on ≥ 3 occasions). Also, compared to this definition, the current study measured BP on only two occasions.

This information has been added as suggested.

Page 8: The section on anthropometrical factors can be renamed as body mass index.

Heading changed to Overweight/obesity as this is the name of the variable in question.

How were weight and height assessed? Was it self-reported or was it measured by the research staff? (On reading the discussion it seems these were self-reported by the parents, but it would be helpful to have this information here). This is important to know since it can affect how the results are interpreted.

The text clearly states in the last paragraph on page 7 that weight and height was measured by a research nurse in the last follow-up and parental reported in previous follow-ups

Page 8: Lines 48-51: It should be BMI calculated as ... and not Obesity and BMI calculated as.....

Changed as suggested.

Page 8: Lines 54-56: Cutoff based on what?

Cut-off are based on the study by Cole referred to, we believe this needs no further explanation as it is a well-known definition.

Page 9: Lines 3-6: Not sure why parental height was included?

Text rewritten to clarify that this refers to child's weight and height at each age.

Page 9: Lines 32-35: Was the definition of borderline high LDL also based on percentiles? If yes, what percentile was used and if no, what was this definition based on?

Percentiles for definition of borderline high LDL added.

Not sure why triglycerides was not included in the measurement for dyslipidaemia. It would have been useful to include this variable.

LDL and HDL were chosen as the most important Cardiovascular risk factors as evidence for these two factors were deemed stronger than for triglycerides.

Bitzur, R., Cohen, H., Kamari, Y., Shaish, A., & Harats, D. (2009). Triglycerides and HDL cholesterol: stars or second leads in diabetes?. *Diabetes care*, 32 Suppl 2(Suppl 2), S373–S377.

doi:10.2337/dc09-S343

Page 9: Socioeconomic variables: The citation given on line 42-43 for the Swedish socioeconomic classification index appears to be incorrect. Though the authors have described that parental occupation was included and maternal education and family income were measured it is unclear how the authors constructed this index in the current study and what the five grades were. Also, from the analysis and results it seems that SES was assessed not only by this index but also from individual SES variables. It would be informative to state that here in this section for clarity. What does each class include and what does it mean for SES?

Clarification regarding construction of the parental occupational class/SEI variable have been added, see page 9.

Page 10: The white collar and blue collar city section could possibly be included in the section on socioeconomic variables and not have a separate section for this variable because white/blue collar cities primarily refer to SES.

Changed as suggested by the reviewer.

Page 10: Potential confounders:

It would be informative to describe how birth weight was assessed – continuous or categorical (the number and description of the categories). However, it is unclear how birthweight is a confounder because it is most probably a mediator. The SES can affect birthweight which can be a risk factor for the cardiovascular risk factors.

Information on birth weight assessment added. It is important to note that the difference between mediator and confounder is where it is believed to sit in the causal path. While we agree that the reviewer could be correct in suggesting birthweight as a mediator we do not think its role is definitively understood either way and is not a central tenet of this work.

Page 10: Statistical analysis:

It is important to describe for each cardiovascular risk factor at which time they were measured. BMI was measured at four time points but it's unclear when the other risk factors were measured. Also, out of the four measurements for BMI which one was used to examine the overall association between SES and overweight/obesity?

We have clarified that all cardiovascular risk factors were measured at the final follow-up as stated in Methods. Clarified that BMI at final follow-up was used in the analysis

It is unclear what the independent variables and potential confounders are. It seems the confounders have been treated as independent variables. It would help to have consistent terms. Based on the research questions/aims of the study the SES variables are the independent variables and cardiovascular risk factors (blood pressure, dyslipidaemias, and overweight) are the dependent variables/outcomes.

To improve consistency in terminology independent variables previously labeled confounder and mediator have been changed to covariates. The primary aim of this study is not to analyze mediation pathways but the relationship between SES and cardiovascular risk factors. However, it was important to show if this relationship remained after adjustment for other important factors, regardless of where those factors might sit in a causal path.

The current study is a prospective cohort study and the outcome is not rare. Therefore it is unclear why logistic regression was used to estimate odds ratios. It would have been more appropriate to compute relative risk or risk ratio.

There is actually no technical problem in utilizing logistic regression in prospective cohort studies as long as the effect size (odds ratio) is interpreted correctly. We expect the reviewer is referring to the known over-estimation of relative risk by odds ratios when the outcome is not rare, say >10% (Knol, M.J., et al., Overestimation of risk ratios by odds ratios in trials and cohort studies: alternatives to logistic regression. Canadian Medical Association journal, 2012. 184(8): p. 895-899). However the aim of this study was not specifically to estimate relative risk but to identify risk factors for inequality outcome and their strength of association with that outcome. The odds ratio is just as valid a measure of effect size as relative risk and we have been very careful to express them as such.

It would be informative to include what multiple imputation method was used and how many datasets were used for imputation.

Multiple imputation was employed as implemented in the SPSS statistics program. A Markov Chain Monte Carlo was employed with predictive models selected according to the measurement scale of each dependent variable specified. Five imputation samples were derived. These details have now been included in the manuscript.

Page 10: Lines 56-60: It is unclear what the four SES variables are (though from table 1 it seems that these are: maternal education, income, SEI, and blue collar/white collar cities). It would be informative to make this clear in the section on socioeconomic variables. It is also unclear what the other independent variables are (though the authors say that the continuous variables are height, age, and birthweight). The variable age was not mentioned in the section on potential confounders. If BMI was one of the outcomes then including height as an independent variable/confounder would be incorrect.

We have specified the SES variables, added information about the age variable. Height was included as it was deemed possible as an important factor especially for high blood pressure and is associated with high SES. BMI is not one of the outcomes (although overweight/obesity is).

Page 11: Lines 6-13: It is interesting to read that effect size was used to determine inclusion of variables in the model. It would be informative to provide a citation as to why the authors decided to use <0.75 or >1.5 as selection criteria to include the variables. Also, on page 10: lines 59-60, the

authors have described that $p \leq 0.05$ or the effect size of $OR < 0.5$ or > 2.0 was used for categorical variables. However, it is unclear in which situation either was used.

The use of effect size criteria, rather than p-values, to include/exclude independent variables was due to limited sample size and consequent low statistical power. We were concerned about omitting clinically important predictive variables due to low power rather than because of low clinical effect. Since regression coefficient or odds ratio, depending on the model, represent the clinical effect size we felt they were a more relevant criterion in this instance. We agree that the thresholds are subjective but do correspond to moderate effect sizes.

Page 11: 29-32: The SES variables are likely to be correlated with each other and there is possibility for multicollinearity that can affect the estimates especially the variance. How did the authors take this into account especially when estimating the odds ratios? Also, it would be informative to list the lifestyle factors because it was not mentioned what these refer to (though it would seem that diet and physical activity). It is unclear why SES should be adjusted for overweight status if this variable is one of the cardiovascular risk factors and hence one of the outcomes/dependent variables.

Examination of the correlations between potentially predictive variables revealed correlations (r 0.23-0.47), as the reviewer suggests, but not to the extent that that collinearity would be expected ($r > 0.7$).

Page 11: Lines 34-46: It would have been more appropriate to state that logistic regression was used to examine the relationship between maternal education at four different time points and overweight/obesity because this is cohort study and not a case-control study. Also, it's unclear why only maternal education was used for this purpose and not other SES variables. The study aim was to determine at what age socioeconomic differences in overweight and obesity occur....The measurements at four time points are likely to be correlated. How did the logistic regression model account for this at each time point? The methodology used does not seem to be satisfactory for answering the question, what age socioeconomic differences in overweight and obesity occur...?

Maternal education was the SES variable with the best prediction of overweight/obesity and was thus chosen as the best SES variable for this part of the analysis.

We have rephrased the research question to indicate the limitations of our analysis (changed from "what age SES differences occurs" to "at what time points during childhood we could identify significant differences". We have not looked at these as "competitors", just sought to identify whether there were ages at which associations were present and subjectively noted that the strength of association increases with age.

It would be informative to examine/describe the characteristics of the cities that were designated as blue and white collar cities and have a descriptive table for it.

Rather than to add an additional table, we have revised the text to more clearly describe the characteristics of these cities and added additional references.

The authors have used ROC to quantify the degree of discrimination. It would be informative to have analysis for calibration also and corresponding calibration plots too.

Our study is primarily concerned with identifying risk factors for inequality and then to quantify how well the identified risk factors discriminate level of outcome, hence the utility of the logistic models and subsequent ROC analysis that we have employed. While calibration has a clear role in diagnostic medicine it would be only of tangential relevance in the context of this study and hence we have not included such analyses. It would of course be possible to do so but we feel the space required to

display the calibration curves would not be justified by the passing reference that would be made to them.

RESULTS

The authors have compared ROCs, however, no statistical testing was done if the differences were statistically significant or not.

The area under the ROC curves for various predictive variables have been compared as the reviewer notes but only in the sense of rank order, and we have been careful to not make claims about whether one variable is a statistically significantly better predictor than another. We do not have specific hypotheses about which variables should be the most predictive and only really need to use the AUCs as a standardized scale on which to rank the discriminative ability of various predictive variables.

It would have been easier if the descriptive statistics were presented first and then the inferential statistics instead of describing the statistics by each variable.

Changed so that descriptive statistics is presented first.

Page 12: Lines 22-28: This information could possibly be deleted.

We decided not to delete this information.

Page 12: Lines 42-48: This information is redundant because these variables are potential confounders and as such not the study aims.

Results on covariates removed.

Page 13: Lines 3-6: It is unclear what indicators of low SES the authors are talking about.

Added; all other SES variables.

Page 13: Lines 23-33: As mentioned above for blood pressure, these variables are not the focus of the research question (they are potential confounders) and hence their estimates are redundant.

Results on covariates removed.

Page 13: The subtitle should possibly be overweight/obese because the authors combined these two into a single category.

Subtitle changed to Overweight/obesity to clarify that it is used as a single category.

Page 14: Lines 3-6: Lifestyle factors (diet, physical activity) were not included in the aims of the study. They are potential confounders.

Results on covariates removed.

Table 2: The study aims relate to SES and cardiovascular risk factors and therefore it would be appropriate to include statistics for these variables only and not for other variables listed in the table.

We believe the reader is interested also in the strength of the association between the covariates and the outcome.

Also, it is unclear why height and overweight should be included as potential confounders when being overweight/obese is an outcome.

Height was included as there is an association with tallness and high SES, this was deemed potentially important primarily on the result of elevated blood pressure but could potentially affect also the result on overweight/obesity, however this was not the case in our study. Although Overweight/obesity was used as an outcome it was also of interest to adjust for overweight/obesity in the associations of SES with elevated blood pressure, HDL and LDL, to see to what extent these associations were attenuated after adjustment.

VERSION 2 – REVIEW

REVIEWER	Yipu Shi Public Health Agency of Canada
REVIEW RETURNED	21-Nov-2019

GENERAL COMMENTS	My questions have been addressed adequately in the revised version.
---

REVIEWER	Rema Ramakrishnan University of Oxford
REVIEW RETURNED	21-Nov-2019

GENERAL COMMENTS	Thank you for addressing the comments. The revised version reads much better. There are some typos and needs few more edits to convey the message more clearly. SPECIFIC COMMENTS FOR RESPONSES Original comment: The justification given here for the study is bit confusing. Based on the information provided here it would have been justifiable to compare the impact of SES on cardiovascular risk factors in the 1980s to that in 2016 and after that. Authors' answer: We believe that this paragraph is a necessary description of the current situation in Sweden, that is increasing income inequality (that some reader might not be aware of) but with a lot of policies that should be protective against health inequalities in children and adolescents Comment: Agree with the response, but could the sentences be reworded to make the information more concise and precise? Currently, the message is still unclear. Original comment: However, it is unclear how birthweight is a confounder because it is most probably a mediator. The SES can affect birthweight which can be a risk factor for the cardiovascular risk factors. Authors' answer:
---

	It is important to note that the difference between mediator and confounder is where it is believed to sit in the causal path. While we agree that the reviewer could be correct in suggesting birthweight as a mediator we do not think its role is definitively understood either way and is not a central tenet of this work. Comment: It is unclear what the authors are trying to say about the bidirectional relationship between parental SES and offspring birthweight. Parental SES can influence birthweight of the offspring, but it is improbable that offspring birthweight can influence parental SES. However, one's own birthweight may influence one's SES through multiple pathways. Also, even though mediation is not the focus of this paper addition of intermediary variables without testing for mediation are likely lead to biased estimates/predictions and therefore should not be included. If mediation analysis was the focus then one would expect direct and indirect (through birthweight) effect of SES. Original comment: The current study is a prospective cohort study and the outcome is not rare. Therefore it is unclear why logistic regression was used to estimate odds ratios. It would have been more appropriate to compute relative risk or risk ratio. Authors' answer: There is actually no technical problem in utilizing logistic regression in prospective cohort studies as long as the effect size (odds ratio) is interpreted correctly. We expect the reviewer is referring to the known over-estimation of relative risk by odds ratios when the outcome is not rare, say >10% (Knol, M.J., et al., Overestimation of risk ratios by odds ratios in trials and cohort studies: alternatives to logistic regression. Canadian Medical Association journal, 2012. 184(8): p. 895-899). However the aim of this study was not specifically to estimate relative risk but to identify risk factors for inequality outcome and their strength of association with that outcome. The odds ratio is just as valid a measure of effect size as relative risk and we have been very careful to express them as such. Comment: Though it would not be appropriate to use odds ratios for this analysis the reviewer believes it is reasonable for this study because the authors were interested in prediction rather than associations and also the ROCs had to be calculated. Perhaps, the aims should be modified to reflect that this was predicting modelling rather than an association study?
--	--

VERSION 2 – AUTHOR RESPONSE

Reviewer(s)' Comments to Author:

Reviewer: 3

SPECIFIC COMMENTS FOR RESPONSES

Original comment:

The justification given here for the study is bit confusing. Based on the information provided here it would have been justifiable to compare the impact of SES on cardiovascular risk factors in the 1980s to that in 2016 and after that.

Authors' answer:

We believe that this paragraph is a necessary description of the current situation in Sweden, that is increasing income inequality (that some reader might not be aware of) but with a lot of policies that should be protective against health inequalities in children and adolescents

Comment:

Agree with the response, but could the sentences be reworded to make the information more concise and precise? Currently, the message is still unclear.

Answer: We have rewritten this part and we believe that the message is now clear for the reader.

Original comment:

However, it is unclear how birthweight is a confounder because it is most probably a mediator. The SES can affect birthweight which can be a risk factor for the cardiovascular risk factors.

Authors' answer:

It is important to note that the difference between mediator and confounder is where it is believed to sit in the causal path. While we agree that the reviewer could be correct in suggesting birthweight as a mediator we do not think its role is definitively understood either way and is not a central tenet of this work.

Comment: It is unclear what the authors are trying to say about the bidirectional relationship between parental SES and offspring birthweight. Parental SES can influence birthweight of the offspring, but it is improbable that offspring birthweight can influence parental SES. However, one's own birthweight may influence one's SES through multiple pathways.

Also, even though mediation is not the focus of this paper addition of intermediary variables without testing for mediation are likely to lead to biased estimates/predictions and therefore should not be included. If mediation analysis was the focus then one would expect direct and indirect (through birthweight) effect of SES.

Answer: Although the relationship between Parental SES > Birthweight > adolescent cardiovascular risk is not clear (for example Parental height confounds this relationship as it affects both birthweight and Parental SES). We have decided to drop Birthweight in this analysis, due to the reviewers' fear that it could lead to biased estimates.

Original comment:

The current study is a prospective cohort study and the outcome is not rare. Therefore it is unclear why logistic regression was used to estimate odds ratios. It would have been more appropriate to compute relative risk or risk ratio.

Authors' answer:

There is actually no technical problem in utilizing logistic regression in prospective cohort studies as long as the effect size (odds ratio) is interpreted correctly. We expect the reviewer is referring to the known over-estimation of relative risk by odds ratios when the outcome is not rare, say >10% (Knol, M.J., et al., Overestimation of risk ratios by odds ratios in trials and cohort studies: alternatives to logistic regression. Canadian Medical Association journal, 2012. 184(8): p. 895-899). However the aim of this study was not specifically to estimate relative risk but to identify risk factors for inequality outcome and their strength of association with that outcome. The odds ratio is just as valid a measure of effect size as relative risk and we have been very careful to express them as such.

Comment:

Though it would not be appropriate to use odds ratios for this analysis the reviewer believes it is reasonable for this study because the authors were interested in prediction rather than associations and also the ROCs had to be calculated. Perhaps, the aims should be modified to reflect that this was predicting modelling rather than an association study?

Answer: We have now clarified in Aims that we investigate whether SES is predictive of Cardiovascular risks rather than associated.

VERSION 3 – REVIEW

REVIEWER	Rema Ramakrishnan University of Oxford
REVIEW RETURNED	25-Dec-2019

GENERAL COMMENTS	Thank you for responding to the comments and revising the paper. There are no major concerns. Following are two minor comments: The new sentence in the introduction, Can we despite these interventions and the relatively low-income inequality, find SES related differences in cardiovascular risk among Swedish adolescents? seems redundant and may be deleted. The sentence, 'Further, to analyze at what ages socioeconomic differences in overweight and obesity occur using longitudinal data from follow-ups in the prospective All Babies In Southeast Sweden (ABIS) study', seems to end abruptly.
---

VERSION 3 – AUTHOR RESPONSE

Answers to Reviewer

Question: The new sentence in the introduction, Can we despite these interventions and the relatively low-income inequality, find SES related differences in cardiovascular risk among Swedish adolescents? seems redundant and may be deleted.

Answer: This sentence has been deleted.

Question: The sentence, 'Further, to analyze at what ages socioeconomic differences in overweight and obesity occur using longitudinal data from follow-ups in the prospective All Babies In Southeast Sweden (ABIS) study', seems to end abruptly.

Answer: Sentence has been rephrased to;

"Further, using longitudinal data we aimed to shed light on at what ages preventive measures might be important for reducing SES differences in overweight and obesity."

We also added two sentences to the Discussion;

"Our results further show that socioeconomic differences in overweight and obesity are not only of importance in adolescents but also early childhood. This finding yields implications for preventive measures and suggests that these measures should start early in life."